# Interspecies Recombination-Led Speciation of a Novel Geminivirus in Pakistan

**DOI:** 10.3390/v14102166

**Published:** 2022-09-30

**Authors:** Aamir Lal, Eui-Joon Kil, Thuy T. B. Vo, I Gusti Ngurah Prabu Wira Sanjaya, Muhammad Amir Qureshi, Bupi Nattanong, Muhammad Ali, Malik Nawaz Shuja, Sukchan Lee

**Affiliations:** 1Department of Integrative Biotechnology, Sungkyunkwan University, Suwon 16419, Korea; 2Department of Plant Medicals, College of Life Sciences, Andong National University, Andong 36729, Korea; 3Agricultural Science and Technology Research Institute, Andong National University, Andong 36729, Korea; 4Department of Life Sciences, School of Science, University of Management and Technology (UMT), Lahore 54770, Pakistan; 5Department of Microbiology, Kohat University of Science and Technology, Kohat 26000, Pakistan

**Keywords:** *Chenopodium album*, recombination, Chenopodium leaf distortion virus, begomovirus

## Abstract

Recombination between isolates of different virus species has been known to be one of the sources of speciation. Weeds serve as mixing vessels for begomoviruses, infecting a wide range of economically important plants, thereby facilitating recombination. *Chenopodium album* is an economically important weed spread worldwide. Here, we present the molecular characterization of a novel recombinant begomovirus identified from *C. album* in Lahore, Pakistan. The complete DNA- A genome of the virus associated with the leaf distortion occurred in the infected *C. album* plants was cloned and sequenced. DNA sequence analysis showed that the nucleotide sequence of the virus shared 93% identity with those of the rose leaf curl virus and the duranta leaf curl virus. Interestingly, this newly identified virus is composed of open reading frames (ORFs) from different origins. Phylogenetic networks and complementary recombination detection methods revealed extensive recombination among the sequences. The infectious clone of the newly detected virus was found to be fully infectious in *C. album* and *Nicotiana benthamiana* as the viral DNA was successfully reconstituted from systemically infected tissues of inoculated plants, thus fulfilling Koch’s postulates. Our study reveals a new speciation of an emergent ssDNA plant virus associated with *C. album* through recombination and therefore, proposed the tentative name ‘Chenopodium leaf distortion virus’ (CLDV).

## 1. Introduction

Interspecific interactions play a primary role in the diversification and organization of life [1]. Numerous recurrent formations of allopolyploid species have been reported in the plant and animal kingdoms [2,3]. Genetic recombination is a major source of genetic variability in viruses and creates new opportunities for the viruses to overcome selection pressures [4,5,6,7]. The expansion of viral host ranges, alteration of transmission vector specificities, and increases in virulence and pathogenesis are associated with recombination [8,9,10]. Recombination between isolates of different species as a source of speciation has been reported widely [11]. Among the DNA viruses, the role of recombination in geminiviruses (family: *Geminiviridae*) in the formation of new DNA virus species is well-documented [12,13,14] and, therefore, plays an essential role in geminivirus diversification and evolution [15,16,17].

The family *Geminiviridae* includes some of the most damaging plant pathogens that affect a wide host range and cause economic losses throughout the world [18,19]. These plant-infecting viruses have very compact monopartite (DNA-A) or bipartite (DNA-A and DNA-B) genomes [20,21,22]. Geminiviruses infect both monocots and dicots [23] and are transmitted by either whiteflies, leafhoppers, treehoppers, or aphids [24,25,26]. Based on the genome organization, host range, and insect vectors, geminiviruses are classified into fourteen genera: *Becurtovirus**, Begomovirus**, Capulavirus**, Citlodavirus**, Curtovirus**, Eragrovirus**, Grablovirus, Maldovirus**, Mastrevirus**, Mulcrilevirus**, Opunvirus**, Topilevirus**, Topocuvirus, Turncurtovirus,* [13] and a few unclassified genera, such as *Olea europaea geminivirus* [27] and *Fraxinus symptomless virus* [28]. Begomoviruses are the largest plant virus genera within the family *Geminiviridae* having single-stranded circular DNA genome, either monopartite or bipartite components, ranging from 2.5 to 3.0 kb in size and are transmitted by whiteflies, i.e., *Bemisia tabaci* [18,24,29]. DNA-A of the New World and Old World begomoviruses contains five and six protein-encoding open reading frames (ORFs), respectively [22]. The DNA-A virion-sense strand encodes a coat protein (*AVI*/*VI*) and a movement protein (*AV2*/*V2,* only in the Old World begomoviruses) [22,30,31], whereas its complementary sense strand encodes a replication initiator protein (*AC1*/*C1*), a transcriptional activator protein (*AC2*/*C2*), a replication enhancer protein (*AC3*/*C3*), and the *AC4*/*C4* protein (*AC4*/*C4*) [32,33,34]. *AC1* (replication-associated protein; Rep) initiates viral DNA replication by binding to iterons within the intergenic region and creates a nick into the conserved TAATATT↓AC sequence [35]. *AC4*, the smallest ORF embedded within the coding region of the Rep protein, is required for monopartite geminivirus infection [36,37]. The DNA-B of bipartite begomoviruses encodes a nuclear shuttle protein (*BV1*) on the virion-sense strand and a movement protein (*BC1*) on the complementary-sense strand [12].

*Chenopodium album* (family: *Amaranthaceae*) is an erect, branched (occasionally unbranched) cosmopolitan weed, which is widely distributed in Canada, Europe, India, Mexico, New Zealand, Pakistan, and South Africa and is ranked among the six most harmful weeds of the world [38]. Except for extreme desert climates, *C. album* is found in all inhabited areas of the world, where it thrives on all soil types and over a wide range of pH [39]. It grows most vigorously in temperate and subtemperate regions; however, it can also potentially affect almost all summer- and winter-sown crops in the tropics and subtropics [40]. *C. album* is responsible for important economic losses in agriculture around the world. In Pakistan, weed species have been identified to be responsible for causing monetary loss worth 3 billion USD annually [41]. Despite some toxic effects, *C. album* is used as a vegetable and a medicinal plant in some regions of the world. A mixed infection of the tomato yellow leaf curl virus (TYLCV) and the tomato yellow leaf curl Sardinia virus (TYLCSV) has been reported to occur in *C. album*. Another begomovirus reported infecting the genus *Chenopodium* is the chenopodium leaf curl virus found in *Chenopodium ambrosiodes* [14]. *C. album* is often used as a herbaceous test plant in plant virology.

In this study, we characterized a new recombinant begomovirus species in *C. album* that has emerged from the begomovirus interspecies recombination events.

## 2. Materials and Methods

### 2.1. Sample Collection and Virus Detection

In June 2018, during a routine survey to record begomoviruses other than the cotton-infecting viruses in Pakistan, leaf distortion was observed in *C. album* weed grown in a residence area, Lahore, Pakistan (Figure 1A,B). Leaf tissues from three plants were collected and stored at −20 °C until processing. Whiteflies, the insect vectors for begomoviruses, were observed in all symptomatic plants. Total nucleic acid was extracted from the samples using the Viral Gene-spin Viral DNA/RNA Extraction Kit (iNtRON Biotechnology, Inc., Seongnam, Korea) following the manufacturer’s instructions. Circular DNA was amplified using the extracted total DNA as a template through rolling circle amplification (RCA) (TempliPhi Amplification Kit; GE Healthcare Life Sciences, Uppsala, Sweden) before being digested with the restriction enzymes *Bam*HI, *Xho*I, *Hind*III, and *Eco*RV (TaKaRa Bio, Inc., Shiga, Japan) [42,43]. All the amplified products digested by all the above-mentioned restriction enzymes were visualized using gel electrophoresis and determined to be approximately 2.7 kb in size. Along with RCA amplification, the presence of begomoviruses was confirmed through PCR amplification of coat protein (CP) and replication enhancer protein (REn) using begomovirus-specific primers, Beg-F (5′-CCGTGCTGCTGCCCCCATTGTCCGCGTCAC-3′) and Beg-R (5′-CTGCCACAACCATGGATTCACGCACAGGG-3′) with target size about 1.1 kb [44]. All these amplicons from both RCA and PCR processing were cloned into the pGEM-3Zf (+) vector (Promega Corporation, WI, USA) and then sequenced by a commercial sequencing service, Macrogen (Seoul, Korea). Sequence contigs were assembled and analyzed using BLASTn and BLASTx [45]. We also attempted to detect satellite molecules, i.e., alphasatellite and betasatellite, and DNA-B through PCR using universal primers [46,47]. Southern hybridization analysis was conducted to confirm the replication of the newly detected virus in the samples using the modified method by Southern et al. [48,49].

### 2.2. Infectious Clone Construction and Infectivity

An infectious clone (1.1 mer) of the detected virus was constructed to check its infectivity in the host plants (Appendix A). Two partial genomes containing restriction sites at the edges, i.e., *Spe*I/*Bam*HI and *Bam*HI/*Xba*I respectively were amplified using primer sets based on the extracted sequence and were ligated into the pGEM-T Easy vector (Promega, Madison, WI, USA), using the TA cloning technique according to the manufacturer’s instructions. This was followed by sequencing (Macrogen, Seoul, Korea) and restriction digestion using specific enzymes. The two partial genomes were introduced into the pCAMBIA1303 vector and first transformed into competent *Escherichia coli* strain DH5α using the heat shock method and then into the GV3101 *Agrobacterium* strains. GV3101 *Agrobacterium* strains (both transformed and untransformed) were cultured in LB broth in the presence of a pCAMBIA1303 selection antibiotic, such as kanamycin (50 mg/L), and strain-specific selection antibiotics, such as gentamycin and rifampicin (50 mg/L), at 28 °C with agitation for 30 h (until the OD value at 600 nm was 0.8–1.0). The untransformed GV3101 *Agrobacterium* with no plasmid was used as the mock. Agro-inoculation was performed by the pin-pricking method [50] in approximately 4- and 6-week-old *N. benthamiana* and *C. album* plants, respectively. Leaf tissue samples were collected from mock and infected plants 28 days post-inoculation (dpi) to check infectivity through PCR processing using the primers: CLDV-IC1/F (5′-ACTAGTTTTGGCAATCGGTGTCTCAC-3′) and CLDV-IC1/R (5′-GGATCCACACTCGTTTACATCC-3′) specifically designed to amplify IC1 (intergenic region, movement protein and coat protein) of the infectious clone of CLDV with a target size of 1.4 kb. The infection caused by CLDV in *N. benthamiana* and *C. album* plants was further confirmed by Southern blot hybridization. 

### 2.3. Target-Specific Primer Construction and PCR Processing

To explore the intriguing genomic composition of the detected virus, i.e., Chenopodium leaf distortion virus (CLDV), primers were designed based on one ORF sequence of the recombinant viruses rather than their identical ORFs in the new virus genomic composition. The ageratum enation virus (AEV) was found to share the highest identity with the Rep and C4 proteins. New primers were designed to target the gene encoding CP of AEV to confirm whether the detected virus was a separate entity. Similarly, primers were designed to target the genes encoding Rep proteins of the rose leaf curl virus (RLCV), duranta leaf curl virus (DLCV), papaya leaf crumple virus (PaLCrV), and catharanthus yellow mosaic virus (CYMV) (Table 1).

### 2.4. Nucleotide Sequences and Phylogenetic Analysis 

A total of 56 full-length DNA-A sequences of AEV (30), RLCV (5), CYMV (6), PaLCrV (9), and DLCV (5) were used in this study including the sequence of CLDV (Appendix A). BLASTn analysis of CLDV showed the sequence identity of these selected viruses though the percentage varies. All sequences were retrieved from the GenBank database (www.ncbi.nlm.nih.gov (accessed on 23 November 2021)) and were aligned at the nicking site in the nonanucleotide motif at the origin of replication (5′-TAATATT//AC-3′). All multiple-sequence alignments were constructed using the MUSCLE method as implemented in MEGA X [51] and manually corrected as well. Phylogenetic tree construction was performed using Mr. Bayes software version 3.2.7a provided by the CIPRES server [52]. In addition, distances were corrected with the best fit model estimated with jModelTest v2.1.6 on XSEDE on the CIPRES Gateway [52,53,54]. Visualization and editing of phylogenetic trees were carried out using a Newick file generated through FigTree in iTOL [55]. The full-length genome sequences of these top hits aligned with the MUSCLE algorithm were subjected to pairwise comparison using SDT v1.2 [56].

### 2.5. Recombination Analysis

Putative recombinants and major and minor “parents” within the datasets were determined using the RDP, GeneConv, Bootscan, MaxChi, Chimaera, SiScan, and 3Seq recombination detection methods implemented on the RDP4 v4.100 suite [57]. In RDP4, the major parent and minor parent are the presumed parent contributing the larger fraction of the sequence and the presumed parent contributing the smaller fraction of the sequence, respectively. Alignments for all methods were performed using default settings and by *p*-value cutoff of 0.05. 

### 2.6. Nucleotide Diversity and Haplotype Variability Indices

The average pairwise number of nucleotide differences per site (nucleotide diversity, π) was estimated using DnaSP version 6.12.03 (Librado and Rozas 2009, Universitat de Barcelona, Barcelona, Spain). DnaSP version 6.12.03 was also used to calculate the number of haplotypes (H), the number of segregating sites (S), haplotype diversity (Hd), testing Tajima D [58], and Fu and Li’s F [59]. Nucleotide diversity estimates the average pairwise differences among sequences while haplotype diversity refers to the frequency and number of haplotypes in the population. Tajima’s D test is based on the differences between the number of segregating sites and the average number of nucleotide differences whereas, Fu and Li’s F test is based on the differences between the number of singletons and the average number of nucleotide differences between every pair of sequences. The statistically significant differences among the mean nucleotide diversity from all datasets were estimated and represented using GraphPad Prism version 8.0. (Harvey Motulsky 1989, Dotmatics, CA, USA).

### 2.7. Estimation of Gene Genealogies through TCS

The method of Templeton, Crandall, and Sing (TCS) resulting genealogical networks identifies both the relationship between the different sequences as well as the number of nucleotide substitutions connecting them. All the sequences of the AEV, RLCV, CYMV, moieties identified in the novel CLDV recombinant begomovirus along with highly identical DLCV and PaLCrV were analyzed using statistical parsimony with the program TCS (v.1.21) implemented in the software Population Analysis with Reticulate Trees (POPART) [60,61]. 

## 3. Results

### 3.1. Detection of a New Virus in C. album

Sequencing of cloned DNA fragments following the digestion of RCA product with *Bam*HI confirmed the presence of a new begomovirus species, i.e., chenopodium leaf distortion virus (CLDV) in all samples (Figure 1C). The complete nucleotide sequence of DNA- A of CLDV was deposited in GenBank under the accession number MN423112. The expected amplicon size of 1.1 kb targeting CP and Ren regions was observed during PCR processing in all symptomatic samples (Figure 1D), yielding the same sequencing results as MN423112. NCBI basic local alignment search tool (BLASTn) analysis revealed a 93% identity of the nucleotide sequence of the newly identified virus (2.7 kb) with that of the rose leaf curl virus (RLCV; MN746285) and the duranta leaf curl virus (DLCV; MN537564) respectively. No satellite molecules or DNA-B were detected in association with the DNA-A using universal primers. Southern blot hybridization assay confirms that the viral DNA was integrated into the genome of the *C. album* by producing a noticeable specific band of the expected full-length genome size (Figure 1E). 

### 3.2. Genome Organization and Homology Analysis of Genes

DNA-A of CLDV contained six ORFs following the Old World organization. Sequence analysis (amino acid level) of each ORF using BLASTX showed that *AC1*, which encodes Rep protein, showed 91% identity with *AC1* of AEV (AGO59951) and *AC4* showed 92% identity with *AC4* of AEV (AGO59954). Similarly, *AC2* (TrAP) and *AC3* (REn) were 99% identical to the *AC2* and *AC3* of the RLCV (QAY29069) and (QAY29070), respectively. The ORF *AVI* (CP) was 100% similar to *AV1* of CYMV (YP_009112873), and *AV2* showed 100% identity with *AV2* of RLCV (ADU33215) (Figure 2A,B). (Detailed comparison of the sequence homology of each ORF of CLDV with the ORFs of other viruses is provided in Appendix A and Appendix A). In the overlapping regions: CP/MP (Δ1) and Rep/TrAP (Δ2), RLCV was found more identical to CLDV than CYMV and AEV respectively (Appendix A). The similarities and differences in the DNA-A of the newly identified virus with the reference sequences are highly noticeable and make a strong case for proving the identity of CLDV as a separate new species.

### 3.3. Infectivity through Infectious Clone Inoculation

Mild symptoms were observed in both CLDV infected plant groups, i.e., *N**. benthamiana* and *C. album*. Leaf tissues were harvested and analyzed by PCR to investigate viral replication ability; the virus was detected in the samples, which confirmed its presence (Figure 3A–E). CLDV (1.4 kb amplicon) was successfully detected through PCR in all three *C. album* samples and three *N. benthamiana* samples. Sample no. 3 of *N. benthamiana* (Lane 6 in Figure 2E) shows a faint band which might be due to poor inoculation or any other experimental error resulting in less virus titer but still, CLDV was reconstituted in this sample when it was sequenced. The virus reconstituted in *C. album* and *N. benthamiana* maintained the exact nucleotide sequence of the original clone. PCR using vector-specific primers showed negative results, which backs the virus detection on its own instead of containing the virus in different parts of the plant. Southern blot hybridization data also confirms the viral infection (Figure 3F).

### 3.4. Target-Specific Primer Construction and PCR Analysis

To explore the intriguing genomic composition of the detected virus, primers were constructed based on one of the ORF sequences of the recombinant viruses rather than their identical ORFs in the new virus genomic composition (Table 1). In all cases, the results remained negative showing the presence of only CLDV in the host sample (Appendix A).

### 3.5. Phylogenetic and Recombination Analysis

To assess the standing evolutionary relatedness among these populations, we performed molecular phylogenetic analysis of CLDV and closely associated viruses, using full-genome sequences (sequences included in the analysis were added in Appendix A). CLDV was found to share a clade with RLCV (Figure 4A). Analysis through Sequence demarcation tool version 1.2 (SDT v1.2.) showed the sequence comparison among the viruses with the revelation of CLDV maximum identity of 93% with RLCV and DLCV isolates (Figure 4B). Recombination analysis detected recombination events in the main genome of CLDV. RDP analysis revealed that CLDV is probably a recombinant genome resulting from a recombination event and originated through recombination between the isolates of the CYMV (GenBank MH643737; 86% similar), AEV (KC795968; 88% similar), and the RLCV (GenBank GQ478342; 93% similar). The recombinant nucleotide coordinates are 2280–1059, spanning the *AC1*, *AC4*, *IR*, *V2*, and *CP* genes. The recombination event was validated by the lower *p*-value of 2.98 × 10^−15^, maximum recombination methods, i.e., RGBMCS3, and an acceptable R score of 0.47 (Figure 5 and Appendix A).

### 3.6. Nucleotide Diversity and Haplotype Variability Indices

We analyzed all datasets comprising AEV (30 sequences), CYMV (6 sequences), DLCV (5 sequences), PaLCrV (9 sequences) and RLCV (5 sequences) to compare the standing molecular diversity. Despite the difference in sample numbers, we were able to calculate average pairwise nucleotide diversities (*π*) for the aforementioned datasets. The average pairwise nucleotide differences were higher for PaLCrV (π = 0.0651), followed by AEV (π = 0.0494), DLCV (π = 0.0442), CYMV (π = 0.0424), and RLCV (π = 0.0257), respectively (Figure 6a). The average number of segregating sites (*θ*_w_) was remarkably higher in the case of PaLCrV (*θ*_w_ = 0.0819), while these values were lower for AEV (*θ_w_* =0.0635), DLCV (*θ*_w_ = 0.0480), CYMV (*θ*_w_ = 0.0344), and RLCV (*θ*_w_ = 0.0255) respectively (Figure 6b). Tajima’s D values for PaLCrV (−1.4063), AEV (−1.1259), DLCV (−0.8438), and RLCV (−0.0109) were found negative whereas CYMV (1.1617) was interestingly found positive (Figure 6c). Likewise, we also found Fu and Li’s F values for each virus: PaLCrV (−1.5289), AEV (−1.3594), DLCV (−0.8903), and RLCV (−0.0024) were found negative whereas CYMV (1.2396) was found positive (Figure 6d and Appendix A).

### 3.7. Estimation of Genealogies through TCS

As the TCS method provides an important tool for dealing with species or genes at the population level and has proved to be a valuable tool in DNA analysis, TCS calculations revealed that most of the isolates sustained a significant number of mutations compared with each other. CLDV is a species that has arisen from the recombination between RLCV, DLCV, and CYMV. Despite CLDV nucleotide sequences having high identity with those of the RLCV, the CLDV genome was found localized in between RLCV and DLCV as shown by the genealogical network analysis (Figure 7).

## 4. Discussion

Geminiviruses have the ability to adapt and evolve quickly as a result of genome-associated changes and recombination events [62,63]. These recombination events have also been documented among members of the genus *Begomovirus* enhancing their virulence [64,65,66,67]. These recombination and recurrent mutations can occur in all plant viruses. In this study, we characterized a new recombinant virus (a name suggested as CLDV) with ORFs originating from three different viruses AEV, RLCV, and CYMV (Figure 2). There are no known begomovirus species with such diverse origins. *C. album* samples with leaf distortion symptoms processed in this study were collected from Lahore, Pakistan from regions where RLCV, DLCV, AEV, and CYMV had been detected previously in various hosts [44,68,69,70]. As all viruses that constitute CLDV were reported in the same region, it compelled us to think of the three possible scenarios of CLDV speciation: (i) collection of all aforementioned viruses from different hosts by whiteflies and transmitted to the *C. album* plant, which might act as a mixing vessel to facilitate interspecific recombination, (ii) these viruses might have been intermixed in any other host and then transmitted to *C. album* by whiteflies, (iii) the interspecies recombination could have been carried out inside the insect vector, which in this case is the whitefly. To investigate the first scenario, we designed the target-specific primers based on one of the ORF sequences of the recombinant viruses rather than their identical ORFs in CLDV (Table 1). PCR amplification showed no positive results which dismisses the possibility of interspecific recombination in CLDV (Appendix A). In the case of the second scenario, we collected a lot of samples i.e., *Vinca rosea*, *Ficus virens*, *Duranta repens*, *Rosa indica*, *Cestrum nocturnum*, etc., from the surrounding areas of the location of CLDV infected *C. album* samples and processed through PCR amplification by using begomovirus specific primers as well as target-specific primers as mentioned above but could not find CLDV in any of the cases (data not shown). There might have been a possibility of missing any other host of CLDV during sample collection. Attempt to detect CLDV from the insect vector, i.e., whiteflies did not succeed either and needs further investigation but the presence of whitefly on *C. album* plants substantiated CLDV as a whitefly vectored begomovirus.

Since genetic variation influences viral emergence, evolution, and vector transmission [4], we investigated the existing genetic diversity of each related virus, i.e., AEV, RLCV, CYMV, PaLCrV, DLCV to determine the extent of genomic variations in these datasets. The average pairwise number of nucleotide differences per site (nucleotide diversity, π), the number of haplotypes (H), the number of segregating sites (S), haplotype diversity (Hd), Tajima’s D value [58], and Fu and Li’s F value [59] were calculated for the aforementioned viruses using DnaSP version 6.12.03 (Librado and Rozas 2009, Universitat de Barcelona, Barcelona, Spain). as shown in Figure 6 and Appendix A. Though we successfully found genetic diversity among these viruses respectively but due to the difference in numbers of sequences (due to the scarcity of viruses in the numbers reported in NCBI GenBank) we cannot conclude that the result data is trustworthy, specifically in regard to CLDV. At least, we observed genetic diversity in all of these CLDV-related viruses which emphasizes the possibility of CLDV existence in the derived class from these viruses.

The possibility of virus existence in the current composition cannot be considered naturally original as clearly recombination events were identified during recombination analysis (Figure 5). Phylogenies are really useful tools to establish genealogical relationships among organisms or their parts (e.g., genes) [60]. Phylogenetic analysis through Mr. Bayes highlighted the evolutionary relatedness among the viruses with the revelation of CLDV localization in between RLCV and DLCV (Figure 4A,B). Along with this traditional method of phylogenetic analysis, an alternative approach TCS [60] was used to provide accurate estimates of gene genealogies at the population level which also showed the same results (Figure 7). On the basis of all these evaluations, we believe that the RLCV is a parent virus here and possesses the *CP* of CYMV and *Rep* and *C4* of AEV through recombination events respectively as shown in Figure 8. Based on the absence of the other component, i.e., the DNA-B, the novel begomovirus CLDV can be considered a monopartite begomovirus with no associated satellite molecules identified and proved through Koch’s postulates by successfully reconstituting the virus from the host after agro-inoculation (Figure 3).

Based on general ICTV demarcation criteria the newly detected virus (CLDV) should be categorized as a new isolate of RLCV (93% sequence identity between them), but the ICTV report clearly demonstrates exceptions in the case of recombinant viruses [29] such as tomato yellow leaf curl Malaga virus and tomato yellow leaf curl Axarquia virus, which have ≥91% identity to both parental viruses (tomato yellow leaf curl virus and tomato yellow leaf curl Sardinia virus) causing both parental species to merge into a single species, despite the fact that all isolates of the parental viruses have <91% identity. Following this rule, CLDV which shows >91% identity to both RLCV and DLCV is categorized as a new species with the proposal of merging these parental species (88% identical) into a single species.

## Figures and Tables

**Figure 1 viruses-14-02166-f001:**
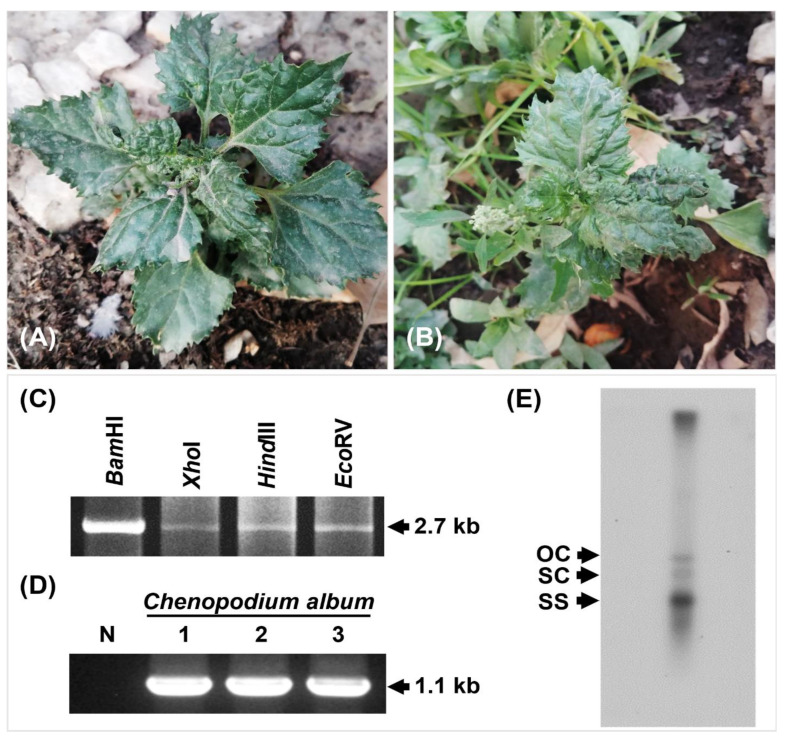
Amplification of virus DNA from the symptomatic *Chenopodium album* plant and confirmation of viral infection; (**A**,**B**) symptomatic samples with leaf distortion symptoms collected and processed. (**C**) Digestion of RCA product using restriction enzymes *Bam*HI, *Xho*I, *Hind*III, and *Eco*RV. (**D**) PCR processing using begomovirus-specific primers of *C. album* samples. N: Negative control, Lane 1–3: *C. album* samples processed. (**E**) Southern blot hybridization of one symptomatic *C. album* sample to confirm virus infection. OC: open circular, SC: supercoiled, SS: single-stranded.

**Figure 2 viruses-14-02166-f002:**
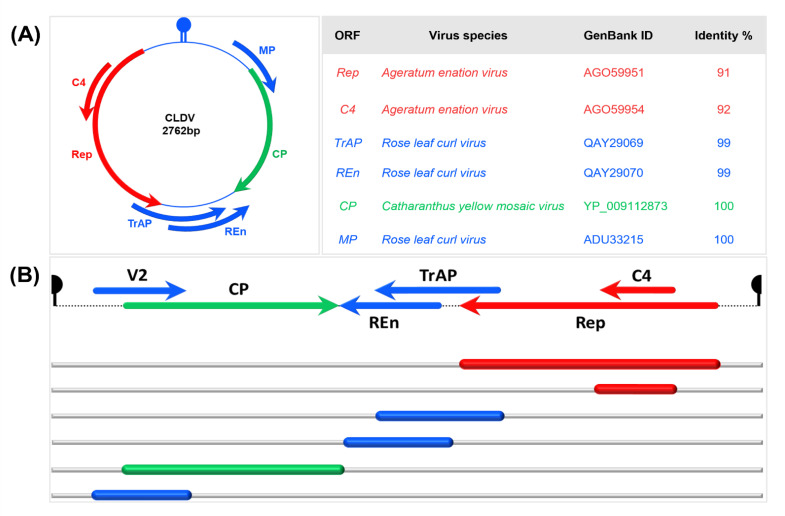
Genome organization of CLDV DNA-A. (**A**) genomic circular map of DNA-A of Chenopodium leaf distortion virus (CLDV) along with open reading frame (ORF) annotation. Each ORF is represented by a specific color-coated line according to its location in the genome. Along with that, the identical ORFs with their GenBank accession numbers and the percentage of identity at the amino acids level have been noted. The genome composition of CLDV depicts ORFs/IR originating from three different viruses, i.e., AEV (Rep, C4), RLCV (TrAP, REn, MP, and IR), and CP (CYMV). (**B**) Linear map of CLDV genome with ORFs representation.

**Figure 3 viruses-14-02166-f003:**
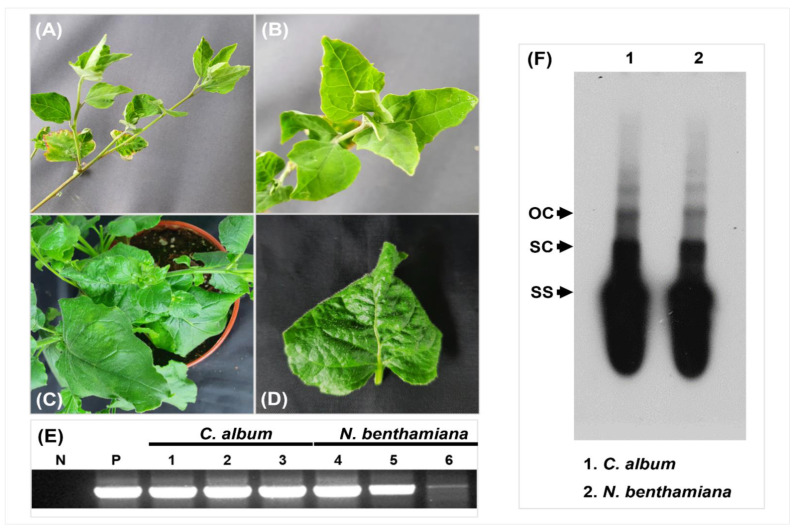
Infectivity of CLDV infectious clones in *C. album* and *N. benthamiana* samples. The results of inoculation assays of CLDV clones observed in (**A**,**B**) *C. album* and (**C**,**D**) *N*. *benthamiana* 28 dpi. Mild symptoms were induced by CLDV in both *C. album* and *N. benthamiana* samples. (**E**) Successful detection of CLDV (1.4 kb amplicon) was done through PCR in all three *C. album* samples and three *N. benthamiana* samples. The mock plant was used as the negative control whereas DNA extracted from one of the infected plants from Pakistan was used as the template for positive control in PCR processing. N: negative control, P: positive control, Lane 1–3: *C. album* samples, Lane 4–6: *N. benthamiana* samples processed. (**F**) Southern blot hybridization data for infection confirmation purpose, Lane 1: *C. album* sample, Lane 2: *N. benthamiana* sample. OC: open circular, SC: supercoiled, SS: single-stranded.

**Figure 4 viruses-14-02166-f004:**
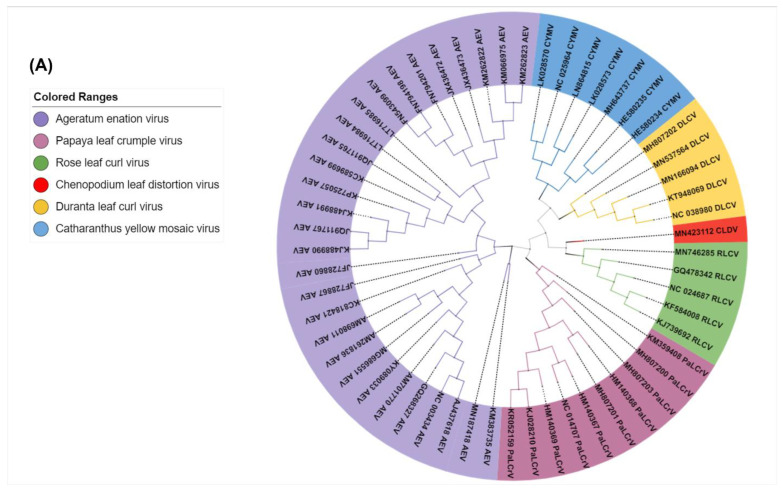
(**A**) Phylogenetic tree generated based on an alignment of the sequences of genomic components (DNA-A) of all viruses with a high percentage identity to CLDV. (**B**) Pairwise complete nucleotide sequence comparison of begomoviruses. Color-coded pairwise identity matrix was generated using the sets of begomovirus sequences by using the SDTv1.2.

**Figure 5 viruses-14-02166-f005:**
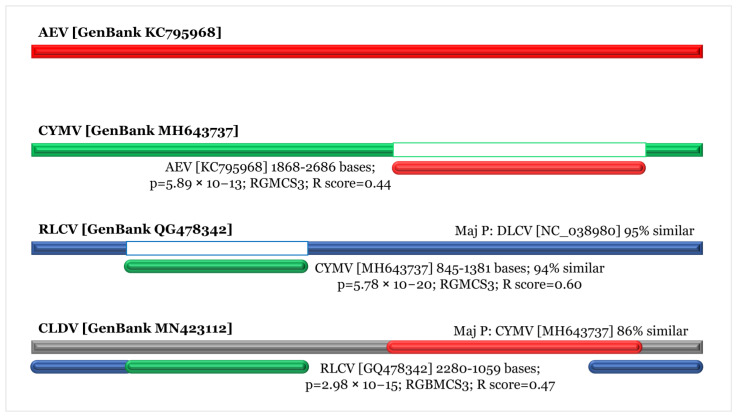
Recombination analysis of recombinant CLDV identified in this study. Recombination events were identified and found between the isolates of the CYMV, AEV, and RLCV.

**Figure 6 viruses-14-02166-f006:**
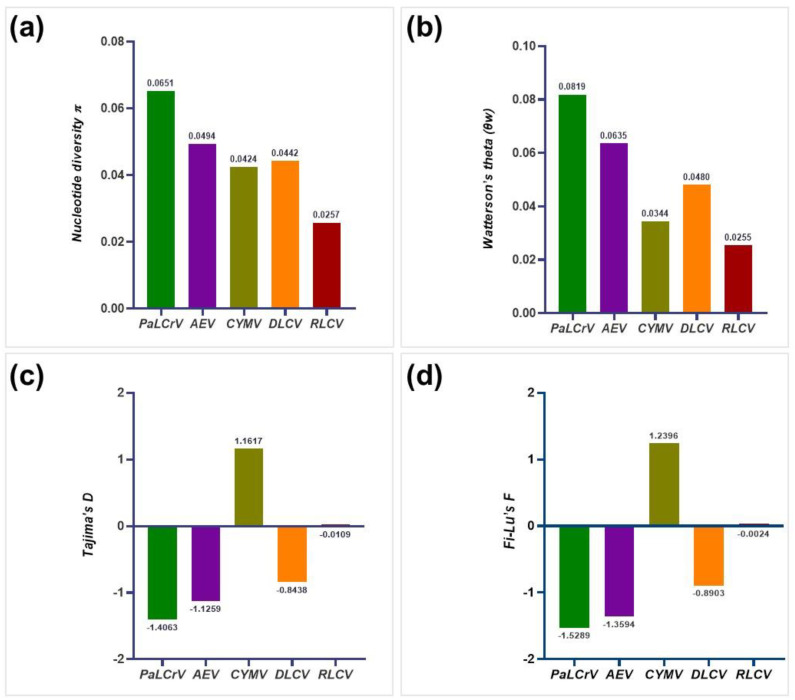
Estimation of genetic diversity was performed for AEV, DLCV, PaLCrV, RLCV, and CYMV populations. The calculated population genetic parameters include (**a**) nucleotide diversity (*π*), (**b**) Watterson’s theta (*θ*_w_), (**c**) Tajima’s D, and (**d**) Fi-Lu’s F.

**Figure 7 viruses-14-02166-f007:**
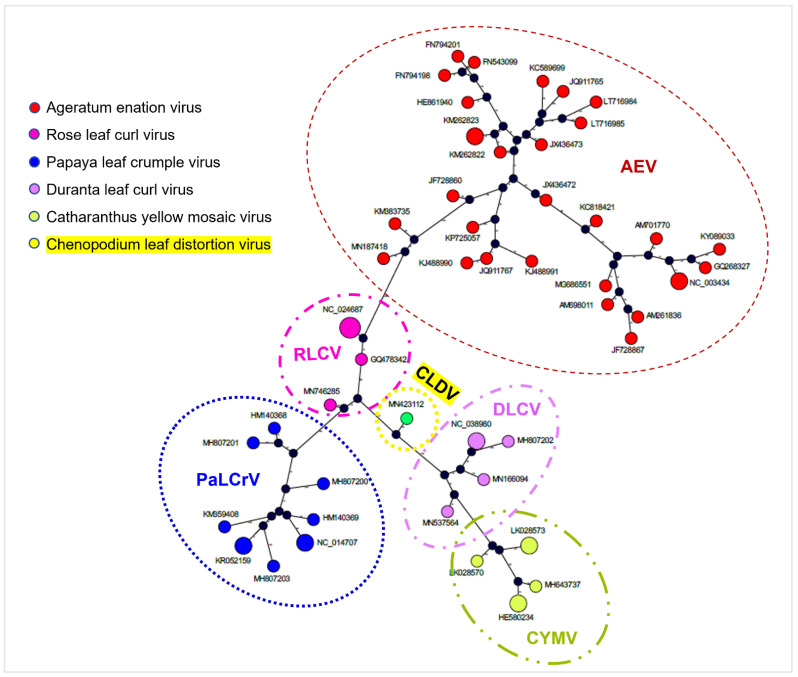
Estimation of gene genealogies through TCS network. CLDV was found located in between RLCV and DLCV.

**Figure 8 viruses-14-02166-f008:**
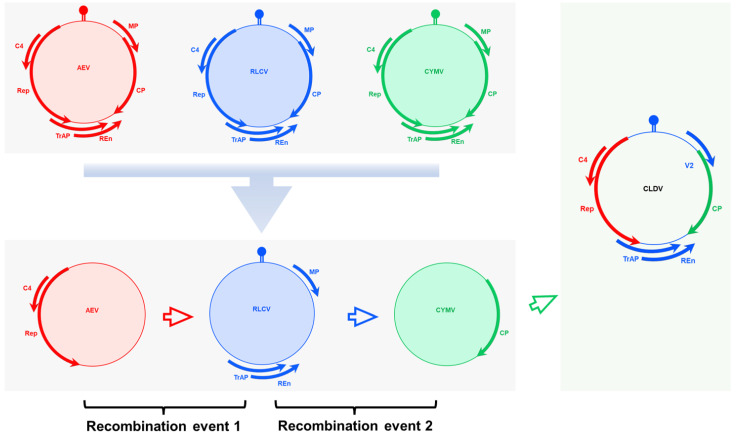
Proposed schematic diagram of CLDV construction through recombination. Two recombination events were hypothesized which results in the existence of CLDV.

**Table 1 viruses-14-02166-t001:** Target-specific primers were constructed to confirm the presence of the viruses in a single host, *C. album*.

Virus	Primer	Sequences 5′-3′	Product Size (bps)	Targeted Area
AEV	AEVCP-F	TGGTCCCCAGACAAACAACT	349	CP
	AEVCP-R	TGGGCTGTCGAAGTTGAGAC		
RLCV	RLVREP-F	GTTCCCTAATGACTCTAAGAGC	384	Rep
	RLVREP-R	AGAAGAAGCCCTCTATCAATTAC		
CYMV	CMREP-F	GCTAAAGCTGCGTCAGCAGA	375	Rep
	CMREP-R	AAAGGAGCAAATGCTCGAACTC		
PaLCrV	PaLRep-F	CAGGATGTACAGGATGTATAGGAG	336	Rep
	PaLRep-R	GTGCTGGGCTCATTATCAAACA		
DLCV	DLCREP-F	TAAAGCTGCTTCAGCTGAACC	365	Rep
	DLCREP-R	GAGCAAATGCTCGAACTCCTTA		

## Data Availability

All data are available and present either in the publication or in the indicated databases.

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
