# Peer review of "Interspecies Recombination-Led Speciation of a Novel Geminivirus in Pakistan"

_viruses, 2022, doi:10.3390/v14102166_

Round 1

Reviewer 1 Report (Previous Reviewer 3)

See attached file

Author Response

This paper is resubmitted to this journal after the first review as an article titled by “Interspecies Recombination-Led Speciation of a Novel Geminivirus in Pakistan” by Lal et al. After incorporation of the reviewers’ comments it is modified/corrected to some extent. However it still has and even more flaws and corrections. The manuscript is not carefully considered yet.

We are grateful to the honourable reviewer for his highly valuable comments and suggestions on our research paper to make it better. We tried our best to respond and address the respected reviewer’s comments one by one in detail.

1) There are no descriptions of the title and legends on all supplementary figures/tables. For example the supplementary Figure 2, no description on asterisk. It means two recombination(s) at least one base upstream (T)/downstream (A) from the first asterisk and one base upstream (G)/downstream (C) at second asterisk or one recombination from (T) and (C). It needs to show the 2nd asterisk should be on the 2nd line over c for readers to check easily and in case of amino acid sequence comparison two asterisks in different color could be better to add and more comments. There seems no check by a native expert in similar fields. From these and other points the manuscript is too premature to submit to this Journal.

  • We added the title and the legends of each supplementary figure and table.
  • We also modified and explained the asterisk as recommended by the respected reviewer.

Note: Line numbers mentioned here in response are according to line numbers used by honorable reviewers in their comments.

Main Comments:

1) As a general it needs to follow the guideline and most recent paper(s) including in italic/non-italic, capital/lower case letters, one space, bold or not, etc...

  • We tried our best to follow the guidelines about italic/non-italic, capital/lower case letters, etc.

2) Regarding the name of restriction enzyme, first three letters should be in italic: e.g. HindIII.They seemed no complete check.

  • We have cross-checked the names of restriction enzymes in the manuscript and have corrected them as suggested by the honorable reviewer.

3) Fig. 2 There is no description/discussion on the mechanism underlined recombination.

  • We are sorry for not understanding well the question/concern of the respected reviewer. We have already tried to describe Figure 2 in section “2. Genome organization and homology analysis of genes”. We also described it in the Discussion part. We hope the respected reviewer will check it and if not good, we need more guidance from the respected reviewer in this regard.
  • The information has also been added in the Figure legend as follows:
    • The genome composition of CLDV depicts ORFs/IR originating from three different viruses i.e., AEV (Rep, C4), RLCV ( TrAP, REn, MP, and IR), and CP (CYMV).

4) Fig. 3 The legend needs much English improvements. In (e) lane 6 is very faint band; there is no explanation on this clone. It needs to explain why this is faint, low accumulation of virus, in relation to virus symptom showing the picture(s), which may be added as supplementary Fig.1? This is very confusing. Fig. 3E should be positioned horizontally instead of vertically (a kind of fundamental error).

  • The figure legend has been revised as follows:
    • Infectivity of CLDV infectious clones in album and N. benthamiana samples. The results of inoculation assays of CLDV clones observed in (A, B) C. album and (C, D) N. benthamiana 28 dpi. Mild symptoms were induced by CLDV in both C. album and N. benthamiana samples. (E) Successful detection of CLDV (1.4 kb amplicon) was done through PCR in all three C. album samples and three N. benthamiana samples. The mock plant was used as the negative control whereas DNA extracted from one of the infected plants from Pakistan was used as the template for positive control in PCR processing. N: negative control, P: positive control, Lane 1-3: C. album samples, Lane 4-6: N. benthamiana samples processed. (F) Southern blot hybridization data for infection confirmation purpose, Lane 1: C. album sample, Lane 2: N. benthamiana sample. OC: open circular, SC: supercoiled, SS: Single-stranded.
  • The detailed information about the faint band in Lane 6 has been added ie section “3. Infectivity through infectious clone inoculation” as follows:
    • CLDV (1.4 kb amplicon) was successfully detected through PCR in all three album samples and three N. benthamiana samples. Sample no. 3 of N. benthamiana (Lane 6 in Figure 2E) shows a faint band which might be due to poor inoculation or any other experimental error resulting in less virus titer but still, CLDV was reconstituted in this sample when got sequenced.
  • The mild symptoms were observed in sample no.3 of benthamiana same as the other two samples of N. benthamiana. That’s why no separate picture was appended to the manuscript.
  • 3E has been positioned horizontally instead of vertically as recommended by the respected reviewer.

5) Fig. 4 Two figs in the same lines are two small to see them in details. Two figs (A) an d (B), shown as mentioned above, should be enlarged as large as possible from the left hand most to the right hand most in the page. One idea is that Fig. 4 A and B should be enlarged in such a way: A in upper half page and B in the bottom half of the same page.

  • Yes, we totally agree with the respected review in this regard, we have enlarged Figure 4 as suggested (A in the upper half page and B in the bottom half of the same page).
  •  

6) Supplementary Fig. 1: Figure title and legend are needed. Also an explanation is needed for bold letters. Brief explanation for the construction is also needed.

  • The Figure title and legend have been added with the explanation of IC construction and Bold letters in the sequence as follows:
    • Schematic diagram of IC construction of CLDV. 1 mer IC was constructed by the addition of restriction enzyme sites i.e., SpeI at the start of IC1 and XbaI at the end of IC2. BamHI is the common point of digestion (end of IC1; start of IC2) existing naturally in the sequence.  The sequences of both IC1 and IC2 have been shown as well in the box on the right side. Restriction enzymes in the sequences (IC1 and IC2) are shown in bold letters. Both IC1 and IC2 are ligated with digested pCambia-1303 followed by the transformation into Agrobacterium strain GV3101.

7) Supplementary Fig. 2: As mentioned above, Figure title and explanation is needed. Explain delta 1 and 2, respectively. The 2nd Asterisk position needs to transfer to the correct position. Also mention here about the structure of recombinant virus.

  • The title and explanation have been added as follows:
    • Analysis of the common region between the ORFs: (Δ1) coat protein-movement protein and (Δ2) replication protein-transcriptional activator protein. The common region of CLDV ORFs was compared with their respective contendents i.e., RLCV and CYMV in the case of Δ1; RLCV and AEV in the case of Δ Analysis was done on both i.e., nucleotide and amino acid levels. In the case of Δ1, RLCV showed more identity with CLDV than CYMV. In the case of Δ2, RLCV showed more identity to CLDV than AEV. The asterisk in red color (*) shows the variations in the nucleotides whereas the asterisk in black color (*) shows the variations in the comparison at the amino acid level.

8) Supplementary Fig. 3: Figure title and legend are needed here. What are these? Why is P absent for three viruses (RLCV, CYMV and PLCV)? More clear pictures are needed for all five viruses.

  • The Figure title and legend have been added as follows:
    • The exploration of the intriguing genomic composition of the newly detected virus (CLDV). The primers were constructed based on one of the ORF sequences of the recombinant viruses rather than their identical ORFs in the new virus genomic composition. (a) RLCV, (b) CYMV, (c) PaLCrV, (d) DLCV, and (e) AEV couldn’t be detected in the infected samples using the specific ORF-based primers. P: positive control (only for DLCV and AEV as didn’t have positive control for RLCV, CYMV, PaLCrV), N: Negative control, Lanes 1-3: infected samples ( album).
  • Unfortunately, we only had the DNA of the DLCV and AEV in our lab which we used as the template for positive control. We didn’t have DNA of RLCV, CYMV, and PaLCrV so no positive control for these viruses during PCR processing. We have changed the dpi of pictures to 600, hopefully, now the pictures will be clearer.

9) Supplementary Table 2: in the bottom description. “: ; ,”(big, medium and small gaps). It needs to change in that way.

  • The bottom description has been amended as recommended as follows:
    • *Present study isolate: AEV; Ageratum enation virus, CLDV; Chenopodium leaf distortion virus, CYMV; Catharanthus yellow mosaic virus,  RLCV; Rose leaf curl virus.

10) Figure 3e and 3f: Figure title and lane number should be added. Some more explanations are needed for the samples and white ladder bands at the end of the sheet. Explain briefly the method.

  • The information has been added as follows (more information have been provided in comment no. 4):
    • Infectivity of CLDV infectious clones in album and N. benthamiana samples. The results of inoculation assays of CLDV clones observed in (A, B) C. album and (C, D) N. benthamiana 28 dpi. Mild symptoms were induced by CLDV in both C. album and N. benthamiana samples. (E) Successful detection of CLDV (1.4 kb amplicon) was done through PCR in all three C. album samples and three N. benthamiana samples. The mock plant was used as the negative control whereas DNA extracted from one of the infected plants from Pakistan was used as the template for positive control in PCR processing. N: negative control, P: positive control, Lane 1-3: C. album samples, Lane 4-6: N. benthamiana samples processed. (F) Southern blot hybridization data for infection confirmation purpose, Lane 1: C. album sample, Lane 2: N. benthamiana sample. OC: open circular, SC: supercoiled, SS: Single-stranded.

11) References: Be careful considering other recent papers. Again all 69 references need corrections. For example, insert “; “ between authors. Single Journal name such as Virology is full name without dot. In case of more than ten authors it needs to add et al. after those authors.

  • We are really grateful to the respected reviewer for our guidance about the references. We have corrected all the references as guided by the respected reviewer.

Reviewer 2 Report (Previous Reviewer 4)

NO more comments

Author Response

<Response to Reviewer no. 2 comments>

L55 five or six instead of five and 6

  • The amendments have been done in Lane 55 as follows:
    • DNA-A of the New World and Old World begomoviruses contains five and six protein-encoding open reading frames (ORFs), respectively [22].

L84 virus v should be not capital letter

  • The amendments have been done in Lane 84 as follows:
    • Sample collection and virus detection

L108 at al. should be italics

  • The et al. has been italicized in Lane 108 as follows:
    • …..to confirm the replication of the newly detected virus in the samples using the modified method by Southern et al. [48, 49].

L110 Fig1 I am not sure these pictures show C. album, it look likes more C. Murale

  • We are really thankful to the respected reviewer to draw our attention to this matter, we did cross-check with our collaborators in Lahore but it is album, not C. murale.

L172 remuve one dots of two at the end of the sentence.

  • One dot has been removed in Lane 172 as follows:
    • ….. the presumed parent contributing the smaller fraction of the sequence, respectively. Alignments for all methods…..

L224 Fig 2 it would be nice to see the sequence homology of the different genes, not only the homology of the recombination origin. I mean please label the identity %, for example in the case of CP 100% with Catharanthus yellow mosaic virus, but what are the identity with Rose leaf curl virus and Ageratum enation virus.

  • Respected Reviewer, we have added the detailed sequence homology of each ORF of CLDV with the viruses as mentioned in the supplementary table 2.

L324 C. album please change to italics

  • I think Lane numbers have been changed. album have been italicized in Lane 335 as follows:
    • ….. viruses might have been intermixed in any other host and then transmitted to album by whiteflies …..

I think Fig 8 is not necessary.

  • We respect the suggestion of our honorable reviewer but we believe that this figure is important to support our hypothesis of the existence of the recombinant virus. It is more explanatory based on the findings so we want to add this figure within the manuscript.

Reviewer 3 Report (New Reviewer)

The manuscript is a correct study discovering a new recombinant Geminivirus species.

Some suggestions:

L55 five or six instead of five and 6

L84 virus v should be not capital letter

L108 at al. should be italics

L110 Fig1 I am not sure these pictures show C. album, it look likes more C. murale

L172 remuve one dots of two at the end of the sentence

L224 Fig2 it would be nice to see the sequence homology of the different genes, not only the homology of the recombination origin. I mean please label the identity %, for example in the case of CP 100% with Catharanthus yellow mosaic virus, but what are the identity with Rose leaf curl virus and Ageratum enation virus

L324 C. album please change to italics

I think Fig 8 is not necessary

Author Response

<Response to Reviewer no. 3 comments>

L55 five or six instead of five and 6

  • The amendments have been done in Lane 55 as follows:
    • DNA-A of the New World and Old World begomoviruses contains five and six protein-encoding open reading frames (ORFs), respectively [22].

L84 virus v should be not capital letter

  • The amendments have been done in Lane 84 as follows:
    • Sample collection and virus detection

L108 at al. should be italics

  • The et al. has been italicized in Lane 108 as follows:
    • …..to confirm the replication of the newly detected virus in the samples using the modified method by Southern et al. [48, 49].

L110 Fig1 I am not sure these pictures show C. album, it look likes more C. Murale

  • We are really thankful to the respected reviewer to draw our attention to this matter, we did cross-check with our collaborators in Lahore but it is album, not C. murale.

L172 remuve one dots of two at the end of the sentence.

  • One dot has been removed in Lane 172 as follows:
    • ….. the presumed parent contributing the smaller fraction of the sequence, respectively. Alignments for all methods…..

L224 Fig 2 it would be nice to see the sequence homology of the different genes, not only the homology of the recombination origin. I mean please label the identity %, for example in the case of CP 100% with Catharanthus yellow mosaic virus, but what are the identity with Rose leaf curl virus and Ageratum enation virus.

  • Respected Reviewer, we have added the detailed sequence homology of each ORF of CLDV with the viruses as mentioned in the supplementary table 2.

L324 C. album please change to italics

  • I think Lane numbers have been changed. album have been italicized in Lane 335 as follows:
    • ….. viruses might have been intermixed in any other host and then transmitted to album by whiteflies …..

I think Fig 8 is not necessary.

  • We respect the suggestion of our honorable reviewer but we believe that this figure is important to support our hypothesis of the existence of the recombinant virus. It is more explanatory based on the findings so we want to add this figure within the manuscript.

Reviewer 4 Report (New Reviewer)

Dear Authors,

I have completed review of the manuscript ID viruses-1836956 entitled "Interspecies recombination-led speciation of a novel geminivirus in Pakistan" submitted by Aamir Lal and collaborators.

The paper contains information on a potentially new member of the family Geminiviridae, genus Begomovirus. The virus, for which the name ‘Chenopodium leaf distortion virus’ was proposed, was isolated from symptomatic plants of a weed specie determined to be Chenopodium album in Lahore, Pakistan. The presence of the virus was confirmed by PCR and Southern hybridization, in addition to sequencing and construction of an infectious clone used for experimental inoculation of Ch. album and N. benthamiana.  Finally, phylogenetic analyzes and recombination events in comparison with related viruses (parental viruses, etc.), as well as genome organization of the newly discovered virus, were presented using various programs and approaches. The major concern related to the article is that it is not apparent from the photographs presented (Figure 1 A and B, Figure 3 A and B) that the host is Chenopodium album (which is crucial for the entire article including the proposed name). The authors are advised to re-examine the particular species and explain on what basis the determination was made - from the photos it looks more like Chenopodium/Chenopodiastrum murale.

Specific comments:

Line 16 – from C. album

Line 19 – replace similarity with identity

Line 42 – family Geminiviridae

Lines 75-80 – the authors are recommended to delete the sentence about the weed situation in Canada (reference 41) and focus on the weed situation in Pakistan (if possible) + in relation to this and the whole article - no need to write virus names with a capital initial (exception: Sardinia) + use normal style (not italic); add sentence about the fact that Ch. album is often used as an herbaceous test plant in plant virology

Line 84 – virus detection

Lines 109-115 – Figure 1 – sections C, D and E including their description should be moved from M&M to Results section

Line 137 – change blue font to black

Line 140 to 148 – change virus names to normal stile + no capital first letter

Line 151 – phylogenetic

Line 152 – remove 0 in case of 05 and 06, 09, 05; DLCV (5) were used

Line 156 – (www.ncbi.nlm.nih.gov) and were aligned…

Line 159 – please specify the reference for MEGA X program

Line 161 – In addition, distances were…

Line 172 – delete one .

Line 176 – Librado and Rozes 2009 – authors are advised to write as software developer (town, country etc.) – same in line 344

Line 186 – GraphPad Prism version 8.0 – producer, country etc. are missing

Line 191 – motives identified

Line 205 – virus names not in italic and without capital letters

Lines 2013-2017 – for most viruses abbreviation is already introduced within the previous text so you can use abbreviations

Figure 2 – be more specific with similarity – aa or nt – also, autors are advised to use the term identity

Line 230 – how do authors explain the development of symptoms on mock-inoculated plants?

Lines 255-257 – The phylogenetic….CIPRES Gateway – more appropriate for M&M section

Line 290 – delete extra space between found and Fu

Line 324 – C. album (should be in italic)

Line 337 – change the text color (currently red)

Author Response

<Response to Reviewer no.4>

The paper contains information on a potentially new member of the family Geminiviridae, genus Begomovirus. The virus, for which the name ‘Chenopodium leaf distortion virus’ was proposed, was isolated from symptomatic plants of a weed specie determined to be Chenopodium album in Lahore, Pakistan. The presence of the virus was confirmed by PCR and Southern hybridization, in addition to sequencing and construction of an infectious clone used for experimental inoculation of Ch. album and N. benthamiana.  Finally, phylogenetic analyzes and recombination events in comparison with related viruses (parental viruses, etc.), as well as genome organization of the newly discovered virus, were presented using various programs and approaches. The major concern related to the article is that it is not apparent from the photographs presented (Figure 1 A and B, Figure 3 A and B) that the host is Chenopodium album (which is crucial for the entire article including the proposed name). The authors are advised to re-examine the particular species and explain on what basis the determination was made - from the photos it looks more like Chenopodium/Chenopodiastrum murale.

  • We are really thankful to the respected reviewer for the valuable comments and specially to draw our attention to this matter, we did cross-check with our collaborators in Lahore but it is album, not C. murale.

Specific comments:

 Line 16 – from C. album

  • Amendment has been done as recommended in Line 16 as follows:
    • Here, we present the molecular characterization of a novel recombinant begomovirus identified from album in Lahore, Pakistan.

Line 19 – replace similarity with identity

  • Similarity has been replaced with identity in Line 19 as follows:
    • DNA sequence analysis showed that the nucleotide sequence of the virus shared 93% identity with those of the rose leaf curl virus and the duranta leaf curl virus.

Line 42 – family Geminiviridae

  • The correction has been made as recommended in Line 42 as follows:
    • The family Geminiviridae includes some of the most damaging plant pathogens that affect a wide host range….

Lines 75-80 – the authors are recommended to delete the sentence about the weed situation in Canada (reference 41) and focus on the weed situation in Pakistan (if possible) + in relation to this and the whole article - no need to write virus names with a capital initial (exception: Sardinia) + use normal style (not italic); add sentence about the fact that Ch. album is often used as an herbaceous test plant in plant virology.

  • The revision has been done as recommended by the respected reviewer in lines 75-80 as follows:
    • In Pakistan, weed species have been identified to be responsible for causing monetary loss worth 3 billion USD annually [41]. Despite some toxic effects, album is used as a vegetable and a medicinal plant in some regions of the world. A mixed infection of the tomato yellow leaf curl virus (TYLCV) and the tomato yellow leaf curl Sardinia virus (TYLCSV) has been reported to occur in C. album. Another begomovirus reported infecting the genus Chenopodium is the chenopodium leaf curl virus found in Chenopodium ambrosiodes [14]. C. album is often used as an herbaceous test plant in plant virology.

Line 84 – virus detection

  • The correction has been made as recommended in Line 84 as follows:
    • Sample collection and virus detection

Lines 109-115 – Figure 1 – sections C, D and E including their description should be moved from M&M to Results section.

  • We understand and respect the suggestion of the honorable reviewer, we have done same previously in the very first draft but one respected reviewer suggested to combine these together just to show detection process in one figure with the samples as number of figures are already bit more.
  • We have described about these sections C, D and E in the result section very well. We look forward to the respected reviewer consent otherwise we’ll change as recommended.

Line 137 – change blue font to black

  • Font color has been changed from blue to black.

Line 140 to 148 – change virus names to normal stile + no capital first letter

  • Virus names have been changed as recommended in lines 140-148 as follows:
    • To explore the intriguing genomic composition of the detected virus i.e., chenopodium leaf distortion virus (CLDV), primers were designed based on one ORF sequence of the recombinant viruses rather than their identical ORFs in the new virus genomic composition. The ageratum enation virus (AEV) was found to share the highest identity with the Rep and C4 proteins. New primers were designed to target the gene encoding CP of AEV to confirm whether the detected virus was a separate entity. Similarly, primers were designed to target the genes encoding Rep proteins of the rose leaf curl virus (RLCV), duranta leaf curl virus (DLCV), papaya leaf crumple virus (PaLCrV), and catharanthus yellow mosaic virus (CYMV) (Table 1).

Line 151 – phylogenetic

  • The correction has been made as recommended in Line 151 as follows:
    • Nucleotide sequences and phylogenetic analysis

Line 152 – remove 0 in case of 05 and 06, 09, 05; DLCV (5) were used

  • The correction has been made as recommended in Line 152 as follows:
    • A total of 56 full-length DNA-A sequences of AEV (30), RLCV (5), CYMV (6), PaLCrV (9), and DLCV (5) were used in this study including the sequence of CLDV (Supplementary Table 1).

Line 156 – (www.ncbi.nlm.nih.gov) and were aligned…

  • The correction has been made as recommended in Line 156 as follows:
    • …. from the GenBank database (ncbi.nlm.nih.gov) and were aligned at the nicking site….

Line 159 – please specify the reference for MEGA X program

  • The reference has been added for MEGA X program.

Line 161 – In addition, distances were…

  • The correction has been made as recommended in Line 161 as follows:
    • …. In addition, distances were corrected with…

Line 172 – delete one .

  • One . has been deleted as follows:
    • …. the smaller fraction of the sequence, respectively. Alignments for all methods …

Line 176 – Librado and Rozes 2009 – authors are advised to write as software developer (town, country etc.) – same in line 344

  • Information has been added in line 176 as follows:
    • …. using DnaSP version 6.12.03 (Librado and Rozas 2009, Universitat de Barcelona, Barcelona, Spain).
  • Same Information has also been added in line 344.

Line 186 – GraphPad Prism version 8.0 – producer, country etc. are missing

  • Information has been added in line 176 as follows:
    • …. using GraphPad Prism version 8.0. (Harvey Motulsky 1989, Dotmatics, California, USA).

Line 191 – motives identified

  • Actually, it is moieties instead of motives. Moieties: each of two parts into which a thing is or can be divided.

Line 205 – virus names not in italic and without capital letters

  • The correction has been made as recommended in Line 205 as follows:
    • …. newly identified virus (2.7 kb) with that of the rose leaf curl virus (RLCV; MN746285) and the duranta leaf curl virus (DLCV; MN537564) respectively.

Lines 2013-2017 – for most viruses abbreviation is already introduced within the previous text so you can use abbreviations.

  • The viruses have been abbreviated as recommended by the respected reviewer.

Figure 2 – be more specific with similarity – aa or nt – also, autors are advised to use the term identity.

  • The amendments have been done as recommended by the respected reviewer.

Line 230 – how do authors explain the development of symptoms on mock-inoculated plants?

  • We are really sorry as we made typing error, there were no symptoms on mock plants. In fact, we were intending to write that mild symptoms were observed in both CLDV infected plant groups i.e., benthamiana and C. album. We have revised the sentence as follows:
    • Mild symptoms were observed in both CLDV infected plant groups i.e., benthamiana and C. album.

Lines 255-257 – The phylogenetic….CIPRES Gateway – more appropriate for M&M section.

  • Lines 255-257 have been removed from the Results section as recommended.

Line 290 – delete extra space between found and Fu

  • Extra space has been removed as follows:
    • Likewise, we also found Fu and Li’s F values….

Line 324 – C. album (should be in italic)

  • album has been italicized.

Line 337 – change the text color (currently red)

  • The color has been corrected as recommended by the respected reviewer.

Round 2

Reviewer 1 Report (Previous Reviewer 3)

separate file

Author Response

Response to Reviewer no. 1 – “- Reviewer 1 – “Fig 4.(a) (b), author response is

  • Yes, we totally agree with the respected review in this

regard, we have enlarged Figure 4 as suggested (A in the upper half page

and B in the bottom half of the same page).

However, the downloaded manuscript showed no change at all.”

  • We have appended the updated figures in zip file last time but couldn’t change it in manuscript as was expecting to get change by the production team.
  • This time we have changed the figures as recommended in the manuscript.

Reviewer 4 Report (New Reviewer)

Please see the comments in the attachment.

Author Response

Response to Reviewer no. 4 – “I have a big doubt that the investigated plant was Chenopodium album. I found the answer that authors checked with colleague not confirmatory enough - it would be nice to describe on which basis determination was done.”

  • We are really grateful to our respected reviewer for his concern about this really important issue. Actually, Chenopodium album and Chenopodium murale are much similar in their looks and confusing to distinguish sometimes as well. This issue has been continuously taken into consideration throughout the sample collection and processing steps. Some of the plant identification apps also showed it as C. murale but the following are the basis of categorization into C. album.
  1. In the samples collection area, we observed mild white powder on the samples. Mostly, album shows these powdery appearances. Though C. murale also shows the same it’s what we mainly observed in the Lahore region.
  2. We confirmed the plant samples at that time with two respected taxonomists, one from Pakistan (our Professor in Taxonomy) and one researcher (taxonomist in RDA) in South Korea. Pakistan. Pakistani professor checked the plant samples by visiting the location at that time and confirmed it as C. album. The researcher in Korea was a bit confused at first but later he also advised us to go as C. album.
  3. Now, when the respected reviewer asked to reconfirm, we contacted our collaborator in Lahore, He visited the site but unfortunately, he couldn’t find any samples there. We contacted our Professor again with these pics and he again confirmed it as album.
  4. We also confirmed the infection in album after inoculating the Infectious clones of CLDV. It is also one supporting argument though we didn’t check its infectivity in C. murale.
  • We are really grateful that the respected reviewer is helping and guiding us to make our manuscript very clear and understandable. We believe that it is C. album but we respect the suggestion and recommendation of the respected reviewer. As we mentioned above our reasons for describing it as C. album if the honorable reviewer still recommends we change it we’ll change it asap.

This manuscript is a resubmission of an earlier submission. The following is a list of the peer review reports and author responses from that submission.

Round 1

Reviewer 1 Report

Line 77:  it will be Tomato yellow leaf curl virus (TYLCV)

Line 78: it will be Tomato yellow leaf curl Sardinia virus (TYLCSV)

Line 97-98: The primes which  have been used for which target sequence and length should be mentioned.

Line 107: The title of the figure 1 should be rewritten as follows:  Amplification of  virus DNA from the sumptomatic  Chenopodium  alum plant and confirmation of viral infection; then…..(a,b)…c…d like that

Line 116: Which restriction enzymes were used and how these two sequences were joined —should be clear. Figure  or map of the infectious clone should be given.

Line 143-145: (The AEV (30), RLCV (05), CYMV (06), 143 PaLCrV (09), DLCV (05) and were used in this study, including sequence of CLDV.  BLASTn analysis of CLDV): full name o the viruses should be written.

Line 160- 162:  (The terms 160 major parent and minor parent are used by RDP4 to refer to sequences that contribute the 161 larger and smaller fractions, respectively, to the recombinant and are regarded as the closest relatives to the true isolates involved in the event.) : should be clear

Line 178: full form of TCS should be given

Line 187-192 (In all symptomatic C. album samples, complete nucleotide sequence i.e., 2,762 bp in length, of a new begomovirus species (CLDV) was obtained by sequencing (GenBank MN423112) cloned through the digestion of the rolling circle amplification (RCA) product with BamHI (Figure 1c). Furthermore, a PCR product of 1.1 kb was obtained using begomovirus-specific primers (Figure 1d), which also confirmed the same sequence of the 191 virus.)—Make it understandable.

Line 217: How the mock inoculation was done? Is it whitefly of sap inoculation. What was the source of inoculum?  No clear

Line 218-219: what are the benthamiana groups or album groups?

Line 248: Based on complete genome analysis, CLCD has max identity of 93% with RLCV and DLCV isolates. That means it may be strain of these two viruses; how it will be a novel virus; however in Line 253, it has shown that CLCD is 95% similar with RLCD. Check it and correct data should be given.

Recombination analysis has not been described clearly; according to RDP4 analysis ad data obtained, it should be described. Numerous publications on data analysis based of RDP4 are available. A table containing data obtained in RDP4 should be furnished.

Figure 5 is ambiguous; there are no requirement of figure showing recombination AEV, CYMV and RLCV

It is suggested that in this MS, there should be included either table 2 or figure 6.

Author is claiming CLCD is a novel virus, thus  repeated test for amplification of more samples, and sequencing of more sequences  are essential.  How may many symptomatic  samples of  C. album were taken in this study are not clear.  In this study,  no DNA-B  and no betasatellite  molecules associated with this disease have not been detected; how come it was possible to produce severe symptoms, ie.,  distortion of leaves of infected C. album as shown in the figure 1a and b.

In silico analysis of the virus  is necessary and important, when the biological characterization of the virus  is complete.  CLCD has been reported as novel virus but complete biological characterization of this virus is lacking Thus incomplete biology of CLCD and huge bioinfotics analysis have no meaning. Is  the virus reported in this MS whitefly transmitted or sap transmitted? Not clear..

Reviewer 2 Report

To the author

The paper well displays the recombinant event;  in certain aspects , I have raised few queries. Add information to those comments.  minor corrections in sentence forming. Otherwise it is a good piece of work. 

Reviewer 3 Report

This paper is an article titled by “Interspecies Recombination-Led Speciation of a Novel                                                                                              2

Geminivirus in Pakistan” by Lal et al. showing a recombinant begomovirus in Chenopodium album, Chenopodium leaf distortion virus (CLDV) and through DNA sequence analyses using complementary recombination detection methods the virus is a recombinant with three different begomovirus species. Similar viral recombination papers have already been published. From this point their results and conclusion are not so highly remarkable. The manuscript is not carefully written in many places including references. There are some flaws and many corrections there. It needs vigorous improvements in English language. Through check by a native expert in similar fields is required before submission to this Journal. From these and other points the manuscript should be re-submitted for this Journal. Main comments are listed below as well as this reviewer-checked pages of the manuscript.

  • As a general it needs to follow the guideline and most recent paper(s) including in italic/non italic, capital/lower case letters, one space, bold or not etc...
  • As a general it needs the full name followed by abbreviation in parenthesis such as AEV in line 135.
  • Regarding the name of restriction enzyme, first three letters should be in italic: e.g. Hin
  • 1 and others: (a) (b) (c) etc. should be in capital as other recent papers.
  • 2 There is no description/discussion on the mechanism underlined recombination. It is curious to know where the border on the nucleotide sequence is. In other words, there are two ORFs overlapped, one is from one virus while the other from other virus species. Describe how they know the boarder by showing a vertical arrowhead.
  • 3 The legend needs much English improvements. In (e) lane 6 is very faint band; there is no explanation on this clone. It needs to explain why this is faint, low accumulation of virus, in relation to virus symptom showing the picture(s), which may be added as supplementary Fig.
  • 4 Two figs in the same lines are two small to see them in details. Two figs (A) an d (B), shown as mentioned above, should be enlarged as large as possible from the left hand most to the right hand most in the page.
  • Table 2: The second horizontal line should be moved by one line between Virus and PalCrV. This line thickness should be different from other top and bottom lines as other papers.
  • 7: Green circle; Green back color needs to be replaced to higher contrast one such as light yellow. Describe the reason why CLDV between RLCV and DLCV instead of CYMV in the text (in line 293).
  • References: All 69 references need corrections.

Reviewer 4 Report

The authors claimed that they identified a new begomovirus which has been experienced an interspecies recombination-led speciation. However when I conducted a BLAST aligment in NCBI with the acc. number of MN423112, I noticed that is is in fact a strain of a Rose leaf curl virus (RLCV) with the number of MN746285 at 93.3% identity and 99% coverage.  It could be also supported from their phylogenetic tree shown in  Fig4. Another key concern is the Koch's role. The plant phenotypes shown in Fig. 1 and Fig. 3 ( infectious clones) are not similar. That means the infectious clones seem not produce an identical phenotypes from the original plant. Whether it means that additional virus presented in the orignal plant, e.g. satellite or another begomovirus? Therefore, let's talk about another minor concern in Fig. 1e. I would like to suggest that they can provide detail primers information in line 195.  Another wrong presentation about the data about Southern blot hybridization assay shown in Fig. 1e. They said that data is to confirm the viral DNA was integrated into the genome of Chenopodium album by producing a noticeable specific band of the expected full-length genome size. It is totally wrong. Please re-write this part.